# Waist-to-hip circumference and waist-to-height ratio could strongly predict glycemic control than body mass index among adult patients with diabetes in Ethiopia: ROC analysis

Abdu Oumer[1]*, Ahmed Ale[2], Zerihun Tariku[1], Aragaw Hamza[3], Legesse Abera[1], Ashenafi Seifu[4]

1 Department of Public Health, College of Medicine and Health Sciences, Dire Dawa University, Dire Dawa, Ethiopia, 2 School of Medicine, College of Medicine and Health Sciences, Dire Dawa University, Dire Dawa, Ethiopia, 3 Department of Anesthesia, College of Medicine and Health Sciences, Dire Dawa University, Dire Dawa, Ethiopia, 4 Department of Anesthesiology, College of Health Sciences, Addis Ababa University, Addis Ababa, Ethiopia

* omab2320@gmail.com

**Data Availability Statement:** All relevant data are within the paper and the supplementary table (S1 Table) included as supplementary file.

## Abstract

### Background

Poorly controlled blood glucose is prevalent and contributes to the huge burden of diabetes related morbidity, and central obesity has a great role in the pathogenesis of diabetes and its adverse complications, which could predict such risks, yet evidence is lacking. Hence, this paper is to evaluate the predictive performance of central obesity indices for glycemic control among adult patients with diabetes in eastern Ethiopia.

### Methods

A survey of 432 randomly chosen patients with diabetes was conducted using a pretested questionnaire supplemented by chart review, anthropometrics, and biomarkers by trained data collectors. The poor glycemic control was assessed using a fasting blood glucose (FBS) level of above 130 and/or an HgA1c level above 7%. Weight, height, waist circumference (WC), and hip circumference (HC) were measured under standard procedures and we calculated waist-to-hip circumference ratio (WHR) and waist-to-height ratio (WHtR). The receiver operating characteristics curve was used to assess the predictive performance of obesity indices for glycemic control using area under the curve (AUC) and corresponding validity measures.

### Results

A total of 432 (92%) patients with diabetes were enrolled with a mean age of 49.6 (±12.4) years. The mean fasting blood glucose level was 189 (±72) mg dl$^{-1}$ where 330 (76.4%) (95% CI: 74.4–78.4%) and 93.3% of them had poor glycemic control based on FBS and HgA1c,

**Funding:** This research was fully funded by Dire Dawa University under the competitive research grant (RAD/011/2013) in 2021/22. The funders had no role in study design, data collection and analysis, the decision to publish, or the preparation of the manuscript.

**Competing interests:** The authors have declared that no competing interests exist.

**Abbreviations:** A/COR, Adjusted/crude/Odds Ratio; AUC, Area under the Curve; BMI, Body Mass Index; CI, Confidence Interval; CVD, Cardiovascular Disease; DALY, Disability Adjusted Life Years; DM, Diabetes Mellitus; FBS, Fasting Blood Sugar; HbA1c, Glycosylated Hemoglobin; HC, Hip Circumference; HIV/AIDS, Human Immune Virus/ Acquired Immune Deficiency Syndrome; NCDs, Non communicable Diseases; ROC, Receiver Operating Characteristic curve; WC, Waist circumference; WHO, World Health Organization; WHR, Waist to Hip Ratio and WHtR-Waist to Height Ratio.

respectively. WC (AUC = 0.90; 95% CI: 0.85–0.95), WHR (AUC = 0.64; 95% CI: 0.43–0.84), and WHtR (AUC = 0.87; 95% CI: 0.83–0.94) have a higher predictive performance for poor glycemic control at cut-off points above 100 cm, 0.95, and 0.62, respectively. However, obesity indices showed a lower predictive performance for poor glycemic control based on FBS. Body mass index (BMI) had a poor predictive performance for poor glycemic control (AUC = 0.26; 95% CI: 0.13–0.40).

## Conclusions

Poor glycemic control is a public health concern and obesity indicators, typically WC, WHR, and WHtR, have a better predictive performance for poor glycemic control than BMI.

## 1. Introduction

Diabetes mellitus (DM) is a chronic metabolic abnormality of glucose metabolism due to insufficient insulin or insulin resistance, characterized by high blood glucose levels [1]. Globally, the incidence of diabetes has increased from 11.3 to 22.9 million over the past three decades, where 476 million people live with DM [2]. In Ethiopia, DM contributes to 4.2% of the non-communicable disease burden among the adult population [3] and more than 5% of overall mortality [4, 5], where more than 75% of deaths occur among adults [6, 7]. On the other hand, obesity especially, central obesity is increasing at alarming rate, where high body mass index (BMI), waist circumference (WC), Waist-to-hip circumference ratio (WHC), and waist-to-height ratio (WHtR) could strongly predict risks of insulin resistance and risks of poor glycemic control [8].

Adherence to the chronic care of DM through anti-diabetic medications and lifestyle modifications is crucial for a better quality of life and treatment prognosis. Hence, having an optimal blood glucose or glycemic control is the critical treatment target, where uncontrolled glycemic control (fasting blood glucose level of above 130 mg dl$^{-1}$ or glycosylated hemoglobin above 7%) is associated with the occurrence of debilitating vascular, nervous, and renal complications for DM patients [1, 9, 10]. Hence, it is imperative to maintain an optimal blood glucose level of below 130 mg dl$^{-1}$, despite the fact that the target might be contextualized for each patient [11]. Although achieving good glycemic control is of a great public health importance, only half of patients with diabetes are achieving glycosylated hemoglobin (A1C) goals (< 7%) for better glycemic control, globally [12, 13].

Poor glycemic control is a major concern for diabetes patients affecting 60.5–65.6% [14, 15]. It is also evident that more than 80% of diabetes could be prevented through the prevention of overweight and obesity. Both are clearly linked to the glycemic control level of diabetic patients [16]. On the other hand, self-monitoring of blood glucose levels is a key aspect of the day-to-day control and follow-up of the blood glucose level. A study showed that lack of a self-monitoring device is significantly associated with poor glycemic control (Adjusted Odds Ratio (AOR) = 3.44; 95% CI = 1.33–8.94) [17].

Hence, the presence of self-monitoring practice or frequent follow up allows patients to monitor the treatment effectiveness against the sated blood glucose target and to take appropriate actions [18, 19]. It is also indicated that more than half of the poor glycemic control could be predicted through routine measurements early [18]. For instance, previous studies showed that obesity indices better diagnose diabetes at higher validity (AUC > 0.80) [20, 21]. Moreover, a higher WHR (AOR = 3.52, 95% CI = 1.23–10.11) increased the risk of poor

glycemic control by four-folds [22]. Furthermore study from other country showed obesity indicators were found to predict risk of poor glycemic control (AUC = 0.58–0.75) in a better way, yet to be confirmed [23].

Poor glycemic control is a major cause of increased cardiovascular, renal, nervous and other complications with a huge economic burden [24]. The World Health Organization (WHO) emphasizes the need for a simple and feasible, yet, valid screening tool to reliably follow their blood glucose level [1]. However, access to the standard FBS or HgA1c measurement is not affordable for the majority of clients, where access to the simple blood glucometer is low. However, Ethiopia has a poor resource for advanced medical care, where laboratory testing especially in rural areas is difficult to implement routinely. Given these circumstances, we sought to see which anthropometric indicators could be used as surrogate markers of glycemic control and we did this in Ethiopia with the aim to validate what has been reported in the West so far.

## 2. Materials and methods

### 2.1. Study setting and study period

This study was conducted in eastern Ethiopia, spanning the Dire Dawa, Ethiosomali, East Harerghe, and Harari regions. The study sites are located an average of half a mile from the capital city of Ethiopia. The study area is ethnically diverse and multicultural, where the variation in diet and lifestyle might affect the risks of obesity and glycemic control level. In the Harari region, an estimated 250, 093 people reside, where 146, 913 are living in urban areas and 122, 942 are males [25]. Dire Dawa has an estimated population of 506,936 as of the 2012 Ethiopian fiscal year, and there are two public and many private hospitals and clinics. The Ethiosomali region had a total population of more than six million over 350,000 square kilometers, making it the second largest region in Ethiopia. The Somali region is located in the east and southeast of the country, with an estimated population of approximately six million people. Based on the 2007 Census, the Zone has a total population of 2,723,850, with 1,383,198 men and 1,340,652 women; with an area of 17,935.40 square kilometers. It has a total of five governmental hospitals in the zone [26]. People in the area mainly rely on industrially processed cereals and whole grain cereals, with a relatively high prevalence of overnutrition in the country. The study was conducted from January to February 2022 for two months.

### 2.2. Study design and populations

A hospital-based cross-sectional study was employed to evaluate the predictive performance of obesity indices to predict poor glycemic control among diabetic patients on anti-diabetic treatment. The source population of this study were all adult patients with diabetes attending hospitals in eastern Ethiopia, while those randomly selected patients with diabetes attending chronic care in the selected hospitals during the data collection period were the study population.

### 2.3. Eligibility criteria

We involved all adult patients with diabetes aged over 18 years who visited the selected hospitals during the study using a random selection procedure. However, those patients with diabetes who were pregnant were excluded from the study. In addition, critically ill patients (like diabetic ketoacidosis, hyperglycemic coma) and those unable to communicate were excluded from the study. Also, clients with serious psychiatric disorders, which make them unable to give oriented responses and unable to communicate, were excluded so as to get accurate and

reliable data. Furthermore, those with abdominal swelling (ascites) secondary to liver, cardiac, kidney, or other causes were excluded from the study. In addition, those with detectable spinal curvature and deformity were not included as height measurement is very difficult and the height estimates from other proxies may not be valid.

## 2.4. Sample size determination

To determine the minimum sample size for this study, a single proportion sample size formula with P as the prevalence of good fasting plasma glucose among patients with diabetes from the previous study (60.5% had poor fasting plasma glucose) from northern Ethiopia [15], 95% confidence level, "Z" critical value at 95% CI and marginal error of "d" of 5% the sample size for the first objective became 367.While the sample size for obesity indices predictive performance was calculated using the binomial variance of AUC (V(AUC)), the critical value at 95% confidence level ($Z_{\alpha/2} = 1.96$) [24], and the required precision ($\epsilon$), as shown in Eq 1.

$$n = \frac{Z^2 1 - \alpha/_2 \, V(AUC)}{\epsilon^2} \tag{1}$$

Where the V(AUC) can be calculated using (0.0099 X $e^{-\alpha/2}$) x (6α + 16) where α = $\phi^{-1}$(AUC) X 1.414 and $\phi^{-1}$ the inverse of standard cumulative normal distribution. There is currently no article evaluating the predictive performance of obesity indicators for glycemic control. Hence, we used a study reporting the prediction for diabetes (AUC = 0.69) by waist-to-hip circumference ratio (WHR) [21]. The required sample size was 223 subjects in each group, for a total of 426. Adding a 10% non-response rate to account for non-response, the final sample size was 469.

## 2.5. Sampling procedures

A stratified two-stage random sampling technique was employed to select patients with diabetes from five selected public health hospitals in eastern Ethiopia. The total sample size was stratified and allocated proportionally to the selected study regions depending on the average 2-month case flows. Then, two hospitals were selected from each study site and the sample size was allocated proportionally to each facility based on the estimated two-month patients with diabetes' flow. Then, the allocated number of patients with diabetes was selected using systematic random sampling at every sampling interval of sample interval (k). The sample fraction for each study site was calculated by dividing the total expected case flow by the sample size allocated at each facility (Ki = Ni/ni).

## 2.6. Variables

The outcome variable of this study was glycemic control ascertained based on FBS and HgA1c measurements done under standard methods. The glycemic control status of the patients was grouped into good and poor fasting plasma glucose. Good fasting plasma glucose is when the FBS is between below 130 mg $dl^{-1}$, while poor fasting plasma glucose is defined as when the FBS is above 130 mg $dl^{-1}$ (hyperglycemia). In addition, based on the HgA1C level, clients were classified into poor glycemic control (HgA1C level of greater than or equal to 7%) and good glycemic control when the HgA1c level is below 7% [12]. Hence, since the FBS and HgA1c are reliable, accurate, and valid measures to assess glycemic control, we considered them as the gold standard tool against which anthropometric measurement of obesity was evaluated for its predictive performance. While the independent variables mainly considered were sociodemographic variables (age, sex, marital status, occupation, and education), anthropometric

measures of obesity (WC, WHR, BMI, and WHtR), the presence of comorbidities, medication adherence, self-management practices, and substance use.

## 2.7. Methods of data collection

Data was collected using a set of structured questionnaires including sociodemographic situations and anthropometric measurements. Primary interviews and patient cards were used to collect data by trained health care workers and graduating health students. Data collectors got the data by interviewing the study subjects directly in their local languages during their facility visit at each health facility. The weight of the subjects was measured using a calibrated electronic weighting scale to the nearest 0.1kg. The clothing and footwear of the clients were kept minimal. Similarly, the height was measured using an adult stadiometer with the client standing with both eyes facing straight ahead, hands on the side, and with their body standing straight. The BMI was then calculated by dividing the weight in kg by the height in meter squared and was expressed in kg $m^2$.

A non-elastic tape meter was used to measure the WC at a point midway between the lowest rib and the iliac crest in a horizontal plane at around the umbilicus while the respondents were instructed to breathe gently out. Waist circumference (WC) was measured midway between the lower rib and the iliac crest on the mid-axillary line. The tape meter was not tightly secured to avoid pressure and bias in measurement. While the HC was measured at the point yielding the maximum circumference over the buttocks or the pelvis. HC at the level of the widest circumference over the great trochanters was measured with a tape meter in a standing position at the end of a gentle expiration. Measurements were made at least twice, and the average of the two measurements was recorded. The blood sample for FBS was collected using a simple glucometer machine using a small capillary blood sample. The glucometer strip was inserted into the glucometer. A safe finger prick was used to obtain a 2ml blood sample for the glucometer strip. The glucometer readings were registered in mg $dl^{-1}$.

## 2.8. Data quality assurance

Pair of trained data collectors were deployed to collect the data from study subjects as anthropometry needs curios measurements. A one-day training was given on appropriate interview techniques, anthropometric measurements like height, WC, HC, and weight, practice before actual data collection. Constructive feedbacks were given for the data collectors by investigators and supervisor until they become clear of the checklist implementation. The intra-observer and inter-observers' technical errors of measurement were calculated after training of the data collectors and supervisors, to measure the reliability of the weight and height anthropometric measurements. Anthropometric reliability assessment was done on 10 study subjects and inter- and interobserver variation were calculated and data collectors with acceptable variations were included for data collection. Cranach's Alpha measure of reliability was used and kappa above 0.7 is acceptable. All standard measuring procedures and instruments were followed while data collection. During data entry in to EpiData the data quality were kept by making legal ranges, skipping patterns, appropriate coding and careful data entry. Pretesting of the tool was conducted among 20 patients with diabetes from private hospitals in Dire Dawa and necessary amendments were made where the question wording, sequences and the number of questions were further modified after pretest.

## 2.9. Methods of data analysis

After being checked for completeness and consistency, the collected data were entered into EpiData Version 3.02 and exported to STATA version 14.0 (StataCorp, College Station, Texas,

USA) and SPSS version 20 for analysis. The data is presented in tables, graphs, percentages, frequencies, means, medians, and standard deviations. After measurement of weight and height, BMI was calculated automatically as weight in kilograms divided by height in meters squared. Similarly, the WHR and WHtR are calculated. A correlation coefficient is reported to assess the relationship between obesity indices and fasting blood glucose.

The outcome variable glycemic control is categorized as "1" and "0" as poor and good glycemic control, respectively. A receiver operating characteristic (ROC) curve was used to evaluate the performance of the obesity indices to predict poor glycemic control. The area under the curve (AUC) is used to assess the predictive power, with a value close to one showing better performance. To specify, the cut off point for poor glycemic control prediction at maximum sensitivity and specificity was estimated using the Youden index with a minimum vertical distance from the upper left side of the ROC, and is estimated at the maximum combination of both sensitivity and specificity (sensitivity + specificity).

## 2.10. Ethical considerations

Ethical clearance was obtained from Dire Dawa University, Institutional Research Ethical Review Board. Written informed consent was obtained from each respondent after explaining the details of the study procedure. The data is to be used only for this research and personal identifications were not recorded. In cases where clients had poorly controlled DM, dietary and medical counseling were given with emphasis on physical activity, healthy dietary habits, and medication adherence. As the data is collected at a facility where their usual chronic follow-up and care for patients with acute complications and poorly controlled diabetes is facilitated, linkage was facilitated for admission and emergency management within the hospital in consultation with the assigned health professionals.

## 3. Results

### 3.1. Sociodemographic characteristics of clients

In this study, a total of 432 adult patients with diabetes (92% response rate) with a mean age of 49.6 (±12.5) years were included. Out of this, the majority of 238 (55.15) were females and were from urban areas (81.9%), respectively. Moreover, 330 (76.4%), 108 (25.8%) were married and employed in governmental organizations. About 119 (27.5%) and 105(24.3%) of respondents attended up to grade 8–12 and at least college education, respectively. Moreover, 160 (38.2%) and 109 (26%) of respondents were not currently working and work at private institutions, respectively (Table 1).

### 3.2. Clinical characteristics of patients with diabetes

With regard to the type of DM, more than three-fourths, 368 (85.2%) of patients were diagnosed with type II DM. Hypertension (38.5%) and tuberculosis (14.0%) were the most commonly reported comorbidities among patients with diabetes while 139 (32.2%) of patients with diabetes did not have any diagnosed comorbidities yet. Furthermore, 31 (7.2%) of patients with diabetes had been diagnosed with peripheral neuropathy. More importantly, only 170 (39.4%) of patients had a glucometer to monitor their own blood glucose levels, which might not be affordable. In addition, almost half, 198 (45.8%) and 187 (43.3%), were being treated by oral hypoglycemic agents and insulin therapy, respectively (Table 2).

**Table 1. Sociodemographic characteristics of DM pts treated in chronic follow up in selected public hospitals in eastern Ethiopia, 2022.**

| Variables | Categories | Frequency | Percent (%) |
|---|---|---|---|
| Sex of the client | Female | 238 | 55.1 |
| | Male | 194 | 44.9 |
| Residence | Rural | 78 | 18.1 |
| | Urban | 354 | 81.9 |
| Marital status (n = 429) | Divorced | 63 | 14.6 |
| | Married | 330 | 76.4 |
| | Single | 27 | 6.3 |
| | Widowed | 9 | 2.1 |
| Educational status (n = 426) | Illiterate | 94 | 21.8 |
| | Primary school | 108 | 25.0 |
| | Grade 8–12 | 119 | 27.5 |
| | College and above | 105 | 24.3 |
| Occupational status (n = 419) | Farmer | 42 | 10 |
| | Government | 108 | 25.8 |
| | Not working | 160 | 38.2 |
| | Self/Private | 109 | 26 |

## 3.3. Lifestyle characteristics and anthropometric parameters

A total of 332 (76.9%) of respondents had the habit of regular physical activity. On the other hand, 110 (25.5%), 24 (5.6%), and 9 (2.1%) of patients with diabetes had the habit of khat chewing, cigarette smoking, and alcohol consumption, respectively. In addition, based on the standard BMI classification, 21 (4.9%), 238 (55.1%), and 173 (40%) of patients with diabetes had malnutrition (BMI < 18.5 kg m$^2$), normal (18.5 < = BMI < 25 kg m$^2$), and overnutrition (BMI > = 25 kg m$^2$), respectively. Furthermore, a total of 357 (82.6%) had a higher WHtR above 0.5 as per the recommended classifications. While 92.8% of women and 76.3% of men had a higher WHC, above 0.85 and 1 for women and men, respectively.

**Table 2. Client morbidity and treatment related factors among patients with diabetes on chronic care at selected public hospitals in Eastern Ethiopia.**

| Variables | | Frequency | Percent (%) |
|---|---|---|---|
| Type of DM | Type 1 diabetes Mellitus | 64 | 14.8 |
| | Type 2 diabetes Mellitus | 368 | 85.2 |
| Medical comorbidity | Hypertension | 166 | 38.4 |
| | Tuberculosis | 61 | 14.0 |
| | HIV/AIDS | 5 | 1.2 |
| | Peripheral neuropathy | 31 | 7.2 |
| | Others* | 32 | 7.0 |
| | No comorbidity | 139 | 32.2 |
| Have glucometer for personnel care | No | 259 | 60.0 |
| | Yes | 170 | 39.4 |
| Mode of treatment | Oral hypoglycemic agents | 198 | 45.8 |
| | Insulin | 187 | 43.3 |
| | Both | 47 | 10.9 |

*Refers to heart diseases, chronic kidney disease, asthma, thyroid abnormality, gall stone, and respiratory illnesses

### 3.4. Magnitude of glycemic control

The average fasting blood glucose level of patients with diabetes was 189 (±72) mg dl$^{-1}$ where 330 (76.4%); 95% CI: 74.4–78.4% of them had uncontrolled blood glucose levels above 130 mg dl$^{-1}$ indicating hyperglycemic and hypoglycemic states, respectively. But, the majority, 326 (75.5%) had an increased blood glucose level while only 4 (0.9%) were in a hypoglycemic state. The prevalence of uncontrolled blood glucose was higher among females (78.2%) compared to females (74%). Clients of urban residents were more likely to have controlled blood glucose levels (24.3%) compared to those who were rural residents (20.5%). Similarly, blood glucose control was better among type II patients with diabetes (24.7% had controlled blood glucose) compared to type I DM (17.2%). Moreover, a relatively better optimal blood glucose target was achieved among those who were currently working and physically active (28%) compared to those who were not currently working (17%). Based on the HgA1c cut-off point of below 7%, 93.3% (95% CI: 91.7–94.9%) had poor glycemic control. While 55% had poor glycemic control with an HgA1c level of above 9% and 38.3% had borderline blood glucose control (HgA1c of 7–9%) (S1 Table).

### 3.5. Predictive performance of anthropometric indices for glycemic control

The validity and predictive performance of anthropometric indices of obesity in predicting glycemic control among patients with diabetes was evaluated as shown below. Generally, anthropometric indices of obesity (WC, HC, WHR, BMI, and WHtR) were found to be less sensitive predictors of glycemic control among adult patients with diabetes in eastern Ethiopia. These indices could slightly predict uncontrolled blood glucose but were not statistically significant predictors. The overall Youden index is below 50%, in that the indices showed a low balance between the sensitivity and specificity in predicting uncontrolled blood glucose among diabetic male and female patients. Moreover, WHtR could better predict uncontrolled blood glucose among males (AUC = 0.48; 95% CI: 0.39–0.57) and females (AUC = 0.47; 95% CI: 0.38–0.56) at an optimal cut-off point above 0.55, with a better specificity in detecting poor glycemic control (70 and 65%) (Table 3).

**Table 3. Predictive performance of anthropometric indices of obesity for glycemic control (based on FBS) among adult patients with diabetes on chronic follow up in selected public health hospitals, eastern Ethiopia.**

| Anthropometric indicators | AUC with 95% CI | Optimal-cut-off | Sensitivity (%) | Specificity (%) | Youden index (%) | SE | p-value |
|---|---|---|---|---|---|---|---|
| Males | | | | | | | |
| WC | 0.50 (0.41–0.58) | 139.5 | 17 | 95 | 12.7 | .045 | 0.908 |
| HC | 0.49 (0.41–0.57) | 122.5 | 22 | 96 | 17.5 | .042 | 0.788 |
| WHR | 0.55 (0.46–0.63) | 1.25 | 15 | 98 | 14.6 | .044 | 0.324 |
| BMI | 0.47 (0.38–0.57) | 24.2 | 39 | 68 | 6.9 | .048 | 0.543 |
| WHtR | 0.48 (0.39–0.57) | 0.55 | 75 | 28 | 3.0 | .046 | 0.651 |
| Females | | | | | | | |
| WC | 0.47 (0.39–0.56) | 134.5 | 15 | 96 | 11.2 | .046 | .566 |
| HC | 0.48 (0.40–0.56) | 104.6 | 46 | 37 | 9.2 | .041 | .646 |
| WHR | 0.52 (0.42–0.61) | 0.93 | 86 | 27 | 12.4 | .048 | .699 |
| BMI | 0.54 (0.45–0.64) | 24.5 | 64.5 | 50 | 14.5 | .047 | .344 |
| WHtR | 0.47 (0.38–0.56) | 0.56 | 76 | 21 | 3.0 | .046 | 0.468 |

*AUC- Area under the curve; WC-Waist circumference; HC-Hip Circumference; WHR-Waist Hip Circumference Ratio; BMI-Body Mass Index; WHtR-Waist to Height Ratio; SE-standard error for the Area Under the Curve (AUC) estimates. The unit for WC and HC is in centimeter, BMI (kg m$^2$), while it is a ratio for others.

**Table 4. Predictive performance of anthropometric indices of obesity for glycemic control (based on HgA1C <7%) among adult patients with diabetes on chronic follow up in selected public health hospitals, eastern Ethiopia.**

| Anthropometric indicators | AUC with 95% CI | Optimal-cut-off | Sensitivity (%) | Specificity (%) | Youden index (%) | SE | p-value |
|---|---|---|---|---|---|---|---|
| WC (cm) | 0.90 (0.85–0.95) | 100 | 78 | 94 | 71 | 0.025 | 0.0001 |
| HC (cm) | 0.76 (0.65–0.86) | 110 | 53 | 99 | 52 | 0.048 | 0.001 |
| WHR | 0.64 (0.43–0.84) | 0.95 | 91 | 44 | 47 | 0.105 | 0.069 |
| BMI | 0.26 (0.13–0.40) | 16.8 [a] | 100 | 0 | 0 | 0.069 | 0.002 |
| WHtR | 0.87 (0.83–0.94) | 0.62 | 70 | 99 | 70 | 0.028 | 0.0001 |

[a] the BMI cut-off point of 16.8 kg m$^2$ could increase the sensitivity to 100% while at a cut-off point of 34.8 kg m$^2$ BMI will be 100% specific, yet not sensitive indicator. This shows the BMI cut-off point shall be optimal in between balancing sensitivity and specificity of the test. Hence, a cut-off point 22.9 kg m$^2$ could be optimally sensitive (64%) and specific (25%) than a higher or lower cutoff point.

The tools were found to be more valid in terms of being more specific than being more sensitive. For instance, WC and HC could be able to diagnose or screen uncontrolled blood glucose (AUC = 0.50; 95% CI: 0.41–0.58); and (AUC = 0.49; 95% CI: 0.41–0.57) at a specificity of 95 and 96%, respectively. WHR at a cut-off point above 1.25 among males and 0.93 among females, could predict poor glycemic control at 98 and 27% sensitivity, respectively. A relatively higher cut-off points for WC (13.5 vs. 134.5 cm for males and females, respectively) was identified for uncontrolled blood glucose at a higher specificity (95% for males and 96% for females). Similarly, HC measurements above 122.5 cm for males and 104.6 for females could be associated with a higher risk of poor glycemic control among adult patients with diabetes, while BMI is found to be better predictive of poor glycemic control among patients with diabetes. For instance, a BMI value above 24.5 for females and 24.1 kg m$^2$ could predict an uncontrolled blood glucose level at a sensitivity level of 50 and 68%, respectively (Table 3).

As clearly indicated in Table 4, WC, WHR, and WHtR were found to be the best predictors of glycemic control based on the optimal HgA1c level. In comparison, BMI has been shown to be less sensitive and specific in predicting glycemic control. For instance, WC measurement above 100 cm (AUC = 0.90: 95% CI: 0.85–0.95) and HC above 110 cm (AUC = 0.76; 95% CI: 0.65–0.86) could be a valid proxy indicator of poor glycemic control among patients with diabetes. More importantly, the WHR has relatively good predictive power (AUC = 0.64; 95% CI: 0.43–0.84) for glycemic control at a value above 0.95 (sensitivity of 91% and specificity of 44%). A WHtR of above 0.62 could be more predictive of glycemic control (AUC = 0.87; 95% CI: 0.83–0.94) at 70% and 99% sensitivity and specificity, respectively (Table 4; Fig 1).

## 4. Discussions

It is known that optimal glycemic control is critical in averting many diabetic-related short and long-term complications for a better quality of life. However, it demands early detection of clinical aberrations through an easy-to-use yet valid and feasible tool. Hence, this study assessed the predictive performance of anthropometric indicators of obesity (WC, WHR, BMI, and WHtR) for glycemic control using standard approaches. Our study showed that about 330 (76.4%); 95% CI: 74.4–78.4 of DM clients had poor fasting blood glucose and 93.3% by HgA1c, which potentially aggravates the burden of DM related complications. The current estimate is also comparable with studies conducted in DareSelam (69.7%) [27], 89.5% in Malaysia [28], and 71% from Jimma [29]. Moreover, a comparable rate of poor glycemic control has been reported from western Oromia (64.9%) [30], and Ambo (50%) [31]. A review paper also showed that the burden of poor glycemic control reaches 66.8% based on FBS levels [14].

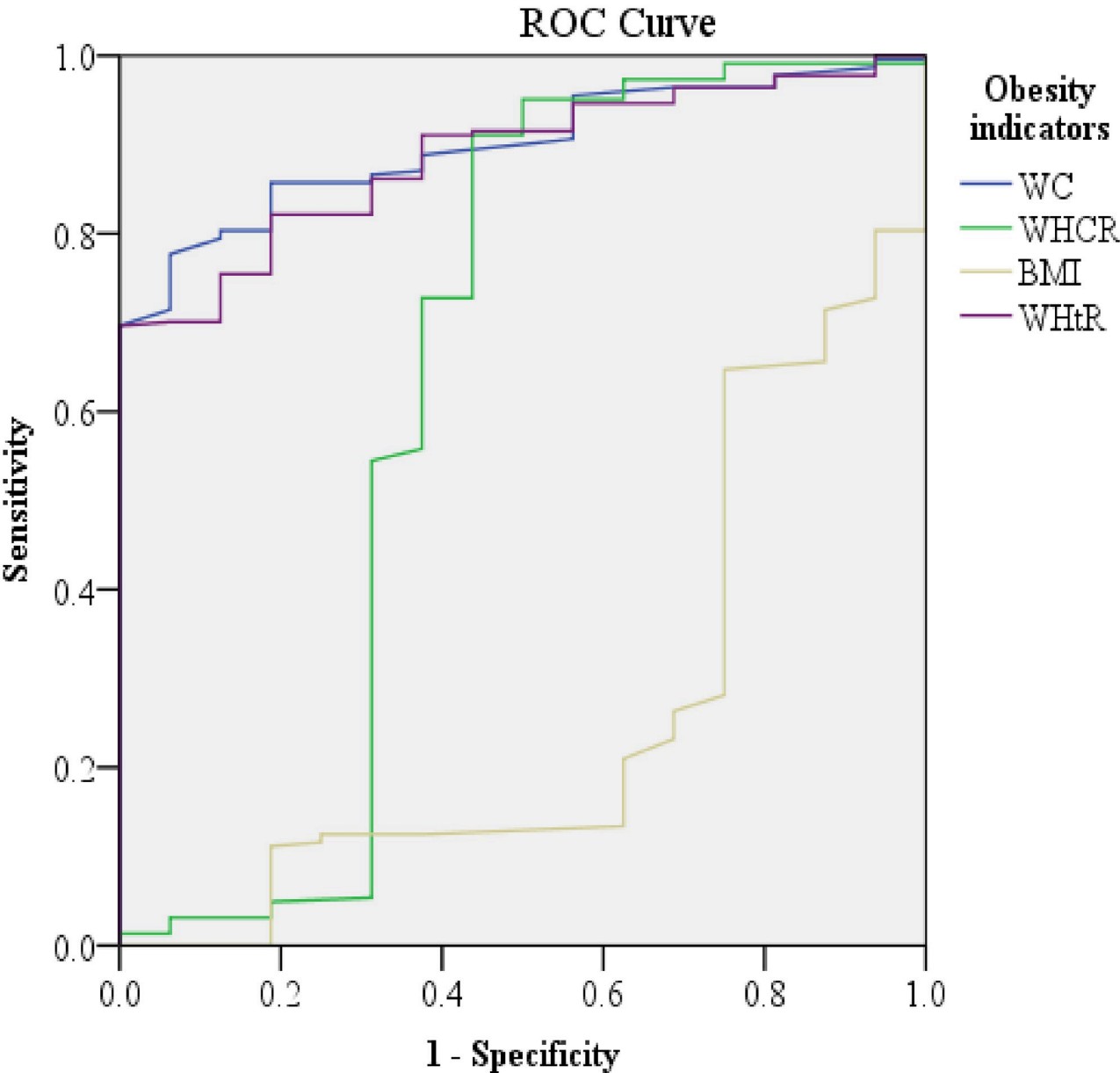

**Fig 1. ROC showing the predictive performance of anthropometric indices of obesity for glycemic control among patients with diabetes on follow up at selected public hospitals in eastern Ethiopia.**

The present analysis indicated that WC measurements above 139.5 and 134.5 cm (AUC = 0.50 for males and 0.47 for females) could predict poor glycemic control among male and female patients with diabetes, respectively. The finding also indicated that the measurement is more specific (95–96%) yet less sensitive. Another similar study among adults showed that WC predicted glycemic control in a better way (AUC = 0.69) [21]. In a country with diabetes mellitus prevalence of 2.0–6.5% [32] and prevailing poor glycemic control [15], increasing obesity might affect treatment outcome. Moreover, a study mentioned central obesity as risk factor for poor glycemic control [33]. Because of the accumulation of visceral fat, WC is a better index for identifying the risk of central obesity and the associated risks of free fatty acid

mobilization and an increased risk of metabolic syndrome [34, 35], and is more predictive of abdominal obesity [36]. These, in turn, place a significant burden on glucose metabolism, leading to increased insulin resistance and poor treatment response. Hence, having a higher cut-off point among those who develop DM (100 cm) could be a more valid cut-off point to predict optimal glycemic control as the target population tends to have a higher WC than healthy individuals. Moreover, WC has a better predictive performance for glycemic control (AUC = 0.90; 0.85–0.95) at a cutoff point above 100 cm based on HgA1c. The test become more specific identifying those with controlled glucose level than the other tests. A lesser cutoff point might allow to capture poor glycemic control in more sensitive ways.

Our findings also indicated that WHR (AUC = 0.55 for males and 0.52 for females) could predict poor glycemic control status. Moreover, a WHR of 1.25 and 0.93 could be used to predict poor glycemic control status at 86% sensitivity and 98% specificity based on FBS. Moreover, WHR was found to predict glycemic control defined by HgA1c (AUC = 0.64; 95% CI: 0.43–0.84) in a better way at a cut-off point above 0.95. A study among healthy adults also found that WHR (AUC = 0.67) better predicts poor glycemic control (AUC > 0.70) [21]. A relatively higher cut-off point is expected to identify poor glycemic control as compared to the usual reference to define high risk for cardiovascular complications (above 1 for males and 0.8 for females) [8, 37]. Hence, it would be imperative to consider a higher WHR as an indicative measure of a higher FBS level and its associated complications. Evidence from South Africa also showed that parameters of central obesity, mainly WC and WHR, are more predictive of glycemic control (p-value below 0.05) at higher AUC [27, 38].

Similarly, BMI could have an optimal predictive performance for poor glycemic control by FBS level (AUC of 0.47 for men and 0.54 for women). It is also evident that higher BMI is strongly correlated with having higher body fat and probably higher WHR and central obesity risks, which ultimately increase the risks of insulin resistance and raised blood sugar levels. However, the relatively low sensitivity of BMI in identifying body fat may limit its use as a diagnostic or screening tool for poor glycemic control when compared to WHR and WHtR [39]. This is mainly due to misclassification of short and lean individuals, where the metabolic risks might be low [40]. However, for the majority of patients higher BMI measurement is indicative of body fatness and risks of poorly controlled blood glucose level. But, the relatively technical nature of the BMI might limit its use by patients as a feasible measure by health professionals at hospitals.

WHtR is another composite indicator comparing the waist circumference of individuals against their height. The higher the weight relative to the height, the higher the risk of obesity and higher FBS due to disturbed blood glucose metabolism. Our analysis also showed that WHtR could be a relatively valid (more specific yet less sensitive) predictor of poor glycemic control among patients with diabetes (AUC of 0.48 for males and 0.47 for females) at a cut-off point of above 0.55. A study among adult Jordanian similarly showed that WHtR (AUC > 0.80) had a better predictive performance for detecting diabetes mellitus compared to WHR and BMI. In addition, WHtR above a cut-off value of 0.6 for women and 0.57 for men could better predict diabetes occurrence among healthy adults [20]. However, a relatively lower cut-off point for WHtR is found for poor glycemic control. WHtR of above 0.42 and 0.51 could allow us to predict poor glycemic control among patients with diabetes with a specificity of 70 and 65%, respectively, based on the FBS definition of glycemic control. Al-Zurfi et al. also demonstrated that a higher WHtR is associated with an uncontrolled glycemic level among patients with diabetes [28]. More importantly, for glycemic control ascertained via HgA1c, the WHtR had a better predictive performance for poor glycemic control (HgA1c > = 7%) at a 0.62 cut-off point (AUC = 0.87; 95% CI: 0.83–0.94).

Despite evidence linking central obesity indicators with the development of diabetic [41, 42], there was a scarcity of evidence to compare with for the prediction of glycemic control. However, a study on the comparative predictive performance of the visceral adiposity index calculated from anthropometric and biomarker information found that it could better predict glycemic control among female patients with diabetes [43]. As the FBS might not indicate the usual blood glucose level as compared to HgA1c, the analysis done via HgA1c is better predictive of overall glycemic control. Based on the promising evidence from the current study and the presence of a biologically plausible explanation linking obesity with glycemic control, anthropometric parameters of central obesity have better predictive power for screening the risks of developing poorly controlled blood glucose levels or poor glycemic control more efficiently.

### 4.1. Limitations of the study

The findings of this study implicated the potential for the use of anthropometric indices of obesity to predict uncontrolled blood glucose. However, the results should be interpreted considering some limitations of the study. A single point FBS measurement might not be representative of the glycemic control status over a period of time. Hence, there is a need for a better and more stable assessment approach. In addition, monitoring the FBS over a period of time could be useful to capture the usual average FBS level or glycemic control status. It should be noted that these parameters are predictive and proxy indices rather than a standard diagnostic tool for poor glycemic control. In addition, due to the low technical feasibility of the HgA1c measurement, the HgA1c was not done on all samples, which might limit its representativeness.

## 5. Conclusion and recommendations

Anthropometric indices of obesity could potentially serve as a relatively valid tool in predicting poor glycemic control among patients with diabetes, yet they are less sensitive in predicting poor glycemic control status. WC, WHR, and WHtR have better predictive performance than HC and BMI measurements in identifying the risks of poorly controlled blood glucose. BMI was found to be a less sensitive and specific tool to predict glycemic control among patients with diabetes. However, anthropometric measurements of obesity with the proposed cut-off points for WC, WHR, and WHtR have a major practical implication in predicting poor glycemic control. Hence, we recommend further study based on a larger representative sample of patients with diabetes with the inclusion of other biomarkers to assess the visceral adiposity index and its predictive ability for glycemic control. In addition, poor glycemic control is more prevalent and patients should follow the recommended medication and lifestyle-related practices to reduce the risk of obesity and ultimately optimal glucose control. Hence, health professionals and patients might use the recommended measures at least to identify imminent risks of poor glycemic control and target their holistic care.

## Supporting information

**S1 Table. The detail dataset for assessing predictive performance of anthropometric indices of obesity for glycemic control in eastern Ethiopia; full dataset.**
(XLSX)

## Acknowledgments

We are grateful to the respective health bureaus, respondents, data collectors, and supervisors for their sincere help and collaboration for the successful completion of the research.

## Author Contributions

**Conceptualization:** Abdu Oumer, Aragaw Hamza, Legesse Abera, Ashenafi Seifu.

**Data curation:** Abdu Oumer, Ashenafi Seifu.

**Formal analysis:** Abdu Oumer, Ahmed Ale.

**Funding acquisition:** Abdu Oumer, Ahmed Ale, Zerihun Tariku, Aragaw Hamza.

**Investigation:** Abdu Oumer, Ahmed Ale, Zerihun Tariku, Legesse Abera.

**Methodology:** Abdu Oumer, Zerihun Tariku, Legesse Abera, Ashenafi Seifu.

**Project administration:** Abdu Oumer, Ahmed Ale, Zerihun Tariku, Aragaw Hamza, Legesse Abera, Ashenafi Seifu.

**Resources:** Abdu Oumer, Ahmed Ale, Zerihun Tariku, Aragaw Hamza, Legesse Abera, Ashenafi Seifu.

**Software:** Abdu Oumer.

**Supervision:** Abdu Oumer, Ahmed Ale, Zerihun Tariku, Aragaw Hamza, Ashenafi Seifu.

**Validation:** Abdu Oumer, Legesse Abera.

**Visualization:** Abdu Oumer, Legesse Abera, Ashenafi Seifu.

**Writing – original draft:** Abdu Oumer, Ashenafi Seifu.

**Writing – review & editing:** Abdu Oumer, Ahmed Ale, Zerihun Tariku, Aragaw Hamza, Legesse Abera, Ashenafi Seifu.

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
