## [Decision Letter · Decision Letter 0]

6 Jul 2022

PONE-D-22-15528Waist-to-hip circumference and waist-to-height ratio could strongly predict glycemic control than body mass index among adult diabetic patients in Ethiopia: ROC analysisPLOS ONE

Dear Dr. Oumer, 

Thank you for submitting your manuscript to PLOS ONE. After careful consideration, we feel that it has merit but does not fully meet PLOS ONE’s publication criteria as it currently stands. Therefore, we invite you to submit a revised version of the manuscript that addresses the points raised during the review process.

We look forward to receiving your revised manuscript.

Kind regards,

Donovan Anthony McGrowder, PhD., MA., MSc

Academic Editor

PLOS ONE

Journal Requirements:

"This research was fully funded by Dire Dawa University under the competitive research grant in 2021/22. The funders had no role in study design, data collection and analysis, the decision to publish, or the preparation of the manuscript. "

"Yes, This research was fully funded by Dire Dawa University under the competitive research grant in 2021/22. The funders had no role in study design, data collection and analysis, the decision to publish, or the preparation of the manuscript. "

Additional Editor Comments:

Dear Dr. Oumer,

The manuscript was revised in accordance with the reviewers’ comments and is provisionally accepted pending final checks for formatting and technical requirements.

Regards,

Dr. Donovan McGrowder (Academic Editor)<o:p></o:p>

Reviewers' comments:

Reviewer's Responses to Questions

**Comments to the Author**

1. Is the manuscript technically sound, and do the data support the conclusions?

Reviewer #1: Yes

Reviewer #2: Partly

2. Has the statistical analysis been performed appropriately and rigorously? 

Reviewer #1: Yes

Reviewer #2: Yes

3. Have the authors made all data underlying the findings in their manuscript fully available?

Reviewer #1: Yes

Reviewer #2: Yes

4. Is the manuscript presented in an intelligible fashion and written in standard English?

Reviewer #1: Yes

Reviewer #2: Yes

5. Review Comments to the Author

Reviewer #1: An interesting point of view - relying on anthropometrics rather than lab tests to diagnose diabetes.

The authors have done a study examining several different measurements of weight / size and glucose control in an area of the world with limited resources. The work is original and the data appear to be correct.

To improve the paper i have several suggestions:

1. Please remove a lot of extraneous information - for example table 1 shows us baseline data but it is never used afterwards to tell us who is or is not overweight or has a high WC.. Hence this info can be removed... Likewise Figure 1 is not needed. The information does not relate to the theme of the paper

2. You do not have to present so much data on sample size determination. Just give the number needed to reach valid calculations.

3. Make the INTRO shorter. Simply state Ethiopia has limited resources for medical care. Laboratory testing especially in rural areas is difficult to do. Given these circumstances you sought to see which anthropometric values could be used as surrogate markers of glucose control. Such simple statements make it clear to the reader what you are looking for. You may also state you are doing this in Ethiopia and trying to validate what has been reported in the West.

4. Consider loosening your criteria for poor control. 130 is actually excellent control for DM. How about looking at a fasting glucose of 150 or 180 or an A1c over 8%?

5. Please focus the DISCUSSION on the cut points and their sensitivity / specificity. Discuss your results and what they mean. Do not pay so much attention to other studies. After all your results are specific to Ethiopia not to other countries.

6. Finally in the DISC you bring up BMI but rarely mention it before. If you are going to include BMI then discuss it in the INTRO so that the reader knows to look for it.

You can basically cut the paper in half in terms of size and lose no information of value if the paper is succinct.

Reviewer #2: Congrats to authors

Kindly to review below comments

The definition of good glycemic control based on FBG (line 151 and other places) has not been a valid point so far as also mentioned as a limitation by the authors. The known international guidelines regarding management of diabetes have set targets for FBG but has not defined a criterion to label “good glycemic control” for a patient based on it. HbA1c and, more recently, time in range for glucose by CGMs are the mostly used descriptions for good diabetes control. Suggest changing the term “good glycemic control” to “good fasting plasma glucose” avoid the gap in the information, unless taking about glycemic control based on HbA1c.

Authors mentioned in line 52 that fasting blood glucose below 70 mg/dl is considered “uncontrolled diabetes”, while this could be true in a sense that glucose is not at target range, but it is not associated with the typical cardiovascular or renal complications of diabetes related to uncontrolled hyperglycemia. Consider removing “below 70” or indicate a different section talking about hypoglycemia specific complications. Line 25 eluded to the same idea, though not described as in line 52, consider also rephrasing line 25.

This allegation of self-monitoring of blood glucose improves diabetes control (mentioned in lines 61-65) is very subjective and depends on many factors, including type of diabetes and therapy used. Even though it is an important tool to monitor diabetes control, the evidence is not yet conclusive. Also reference 9 is for type 1 diabetes which are very specific group of patients, cannot use it to generalize it to all patients with diabetes. Consider rephrasing or removing it. The same paragraph repeated in lines 76-78.

Suggest rephrasing “DM patients” to “patients with DM” in the following lines: Lines 101, 106, 107, 138, 142, 229, 242 (twice), 267, 428, 431, 434

Suggest rephrasing “diabetes patients” to “patients with diabetes”: Lines 21, 23, 102, 103

Suggest rephrasing “diabetic patients” to “patients with diabetes”: Lines 282, 299, 300, 315

Below more specific points to consider

• Line 18: The word poorly should start with a capital letter. Also, the sentence is missing an end, a huge burden of what?

• Line 28: The word “Receiver” should start with a small letter

• Line 29: Add AUC initials following “area under the curve”

• Line 31: Not well written in the part (where 300), suggest rewrite

• Line 33: What are “WHR” and “WHtR”, those initials were never described in the previous text

• Line 38: The word “poor” should start with a capital letter

• Line 40: The word “anthropometric” should start with a capital letter

• What happened to reference 4?

• Line 59: To mention that this data is in Ethiopia

• Line 61: Suggest rephrasing “diabetic patients” to “patients with diabetes”

• Lines 66-68: More points about glucose self-monitoring even though the manuscript is concerning a very different topic.

• Initials AOR have not been described in the manuscript

• Lines 88-98: Much information mentioned here about census and populations, but no references mentioned.

• Line 107: The word “study” possibly missing a following word “period”

• Lines 107-109: It is very sufficient and understandable to mention that patients with DM who were pregnant, were excluded without further explanation.

• Sample size determination: section is very detailed, summary would be best

• Ethical considerations: section is very detailed, summary would be best

• Line 228: Consider removing “of clients”

• Line 231: Add respectively at the end of the sentence.

• Line 232: Re-write the value “105924.3%”

• Line 239: Consider removing “of DM patients”

• Line 242: Change “where” to “while”

• Line 260-261: The terms “undernourished” and “overnourished” seems less scientific terms, consider changing them.

• The two figures are not clear, quality id not optimal

• Line 267: Not well written in the part (where 330), suggest rewrite

• Magnitude of glycemic control: not clear from the text if the differences between associated values were significant or not

• Table 4 has not been labeled and seems attached to table 3

• Line 336: Consider changing “DM clients” to “patients with diabetes”

• Line 347: Not understandable sentence, rephrase

• Discussion: Three pages long is extensive for a discussion, recommend shortening

---

## [Author Response · Author response to Decision Letter 0]

18 Jul 2022

From authors

To: Plos One editorial

Response to reviewers and editor’s comments

Please note that the funding statement placed in the online system is correct and we removed it from the manuscript. Furthermore, we removed the funder from the acknowledgment section. The updated funding statement in the online system “This research was fully funded by Dire Dawa University under the competitive research grant (RAD/011/2013) in 2021/22. The funders had no role in study design, data collection and analysis, the decision to publish, or the preparation of the manuscript.” Is correct.

We put the same funding information in the online and within the revised manuscript.

Regarding the data availability statement 

Since the data include data of patients with diabetes and medical history of patients who had HIV/AIDS and other more ethnically sensitive datasets prevent us from sharing the data publicly. However, the parts of the datasets could be shared to the concerned body up on reasonable request to the corresponding author. Hence, we shared the dataset as “S1 Table” with the current submission and updated the statement accordingly. We also cited the dataset in the data availability as

All relevant data are within the manuscript and the supplementary table (S1 Table) attached as supplementary file. 

Thank you very much

---

## [Editor Report · Decision Letter 1]

16 Aug 2022

Waist-to-hip circumference and waist-to-height ratio could strongly predict glycemic control than body mass index among adult patients with diabetes in Ethiopia: ROC analysis

PONE-D-22-15528R1

Dear Dr. Oumer,

We’re pleased to inform you that your manuscript has been judged scientifically suitable for publication and will be formally accepted for publication once it meets all outstanding technical requirements.

Kind regards,

Donovan Anthony McGrowder, PhD., MA., MSc

Academic Editor

PLOS ONE

Dear Dr.Oumer,

The manuscript was revised in accordance with the reviewers’ comments and is provisionally accepted pending final checks for formatting and technical requirements.

Regards,

Dr. Donovan McGrowder (Academic Editor)<o:p></o:p>

---

## [Editor Report · Acceptance letter]

5 Oct 2022

PONE-D-22-15528R1 

Waist-to-hip circumference and waist-to-height ratio could strongly predict glycemic control than body mass index among adult patients with diabetes in Ethiopia: ROC analysis 

Dear Dr. Oumer:

I'm pleased to inform you that your manuscript has been deemed suitable for publication in PLOS ONE. Congratulations! Your manuscript is now with our production department. 

Kind regards, 

on behalf of

Dr. Donovan Anthony McGrowder 

Academic Editor

PLOS ONE